# Estimation of Energy Consumption and Flight Time Margin for a UAV Mission Based on Fuzzy Systems

Luis H. Manjarrez [1], Julio C. Ramos-Fernández [2], Eduardo S. Espinoza [1,3,*] and Rogelio Lozano [1,4]

1 Center for Research and Advanced Studies of the National Polytechnic Institute, Mexico City 07360, Mexico
2 Department of Mechatronics Engineering, Polytechnic University of Pachuca, Hidalgo 43830, Mexico
3 National Council for Science and Technology, Mexico City 07360, Mexico
4 HEUDIASYC CNRS, Université de Technologie de Compiègne, 60319 Compiegne, France
* Correspondence: eduardo.espinoza@cinvestav.mx or eespinoza@conacyt.mx;
Tel.: +52-5557-47-3800 (ext. 4263)

**Abstract:** An essential aspect to achieving safety with a UAV is that it operates within the limits of its capabilities, the available flight time being a key aspect when planning and executing a mission. The flight time will depend on the relationship between the available energy and the energy required by the UAV to complete the mission. This paper addresses the problem of estimating the energy required to perform a mission, for which a fuzzy Takagi–Sugeno system was implemented, whose premises were developed using fuzzy C-means to estimate the power required in the different stages of the mission. The parameters used in the fuzzy C-means algorithm were optimized using particle swarm optimization. On the other hand, an equivalent circuit model of a battery was used, for which fuzzy modeling was employed to determine the relationship between the open-circuit voltage and the state of charge of the battery, which in conjunction with an extended Kalman filter allows determining the battery charge. In addition, we developed a methodology to determine the minimum allowable battery charge level. From this, it is possible to determine the available flight time at the end of a mission defined as the flight time margin. In order to evaluate the developed methodology, a physical experiment was performed using an hexarotor UAV obtaining a maximum prediction error equivalent to the energy required to operate for 7 s, which corresponds to 2% of the total mission time.

**Keywords:** SoC estimation; fuzzy clustering; multirotor UAV

## 1. Introduction

UAVs are a booming technology, since they represent a versatile platform for a wide range of applications. This technology has found wide acceptance in the energy, construction, and agriculture industries, where it is mainly used for mapping, inspection, photography, and filming. This situation has led to the global drone market being valued at USD 30.6 billion in 2022, and it is estimated that this could reach USD 55.8 billion by 2030 [1]. However, these platforms are susceptible to emerging risks due to technical and operational issues, such as environmental factors, tampering, technical failures, and even cyber-attacks [2,3].

If we combine the continuing growth in the use of these platforms with the risks associated with their operation, it is evident that establishing safety measures in their operations is critical. This is reflected in the rules and regulations adopted worldwide, which limit the types of vehicles, along with the allowed flying zones and operating conditions [4,5].

One of the key factors to guarantee integrity during an operation is that the assigned tasks are aligned with the capabilities of the vehicle. The maximum reachable flight time is an essential parameter since, in order to establish a safe mission profile, in addition to the determination of the time required to complete every mission stage, a safe energy margin must be considered to allow operating for an additional amount of time to successfully

complete it. This represents a safety measure against variations in energy consumption or situations not contemplated that could compromise the aircraft's integrity, people's security, and the environment in which the mission is carried out.

Multirotor UAVs are mostly battery-powered, and therefore, their maximum flight time depends on the available battery energy and the discharge rate. In turn, as stated in [6], the discharge rate will depend on many factors, such as:

- Vehicle design: aerodynamic design, weight, number of actuators, avionics, and energy efficiency.
- Operating environment: air density, wind speed, relative wind direction.
- Dynamics: speed, acceleration, and direction of motion.
- Mission: payload and area of operation.

Furthermore, there are other factors that can affect the energy consumption of the system, such as rotor and hardware failures. In this scenario, the remaining faultless rotors are forced to operate in a region of lower energy efficiency [7], reducing the available energy of the battery due to saturation phenomena in the actuators. Therefore, the information provided by the manufacturers, or that obtained from a performance test under specific conditions, should only be considered as a reference when a mission profile is established.

In this sense, predicting the behavior of the discharge rate and the available energy in a battery makes it possible to know whether the planned mission can be completed successfully, and even to anticipate whether or not it can be completed under conditions that cause unforeseen changes in consumption.

There are two main ways of estimating the energy required to complete the trajectory: (i) using mathematical models that employ the physical characteristics of the vehicle, and its operating speed [8,9]; or (ii) using empirical models employing regressions for a predefined data set [10]. However, such techniques do not consider possible fluctuations in consumption that could affect the capacity to complete a mission successfully.

Fuzzy systems have been shown to be suitable for managing energy-related aspects of UAVs, as can be seen in [11], where a fuzzy system in conjunction with the PSO algorithm was employed to manage the power supply of a hybrid-powered system, showing favorable results in fuel economy while maintaining robustness to variations in power consumption variation. In addition, fuzzy systems have been employed in other UAV-related tasks, such as in control [12] and decision making during the mission [13]; however, these systems have not been used for the calculation of required energy during a mission.

In order to provide a solution for energy estimation in a multirotor UAV, so that it can operate under persistent changes in the energy requirement, we developed a fuzzy-based methodology to determine the total energy required to complete a specific mission based on the vertical and horizontal velocities, the period during which it travels at those velocities, and the power-estimation error for a given state of the UAV.

The proposed methodology is based on fuzzy systems, which are some of the empirical methods. The use of Takagi–Sugeno fuzzy systems was due to their ability to recreate with adequate accuracy the existing functions among the parameters affecting energy consumption. In addition, the structure of the method allows it to be extended to include other factors that affect the energy required without major modifications to the structure. Unlike to the works presented in the literature, it has been conceived for use during the execution of the mission, and not only as a way of estimating the energy required a priori. This provides an important advantage for the safe operation of UAVs, since it not only allows one to know in advance if the mission to be performed is feasible, but also, once it is in progress, it allows one to anticipate variations in consumption that could jeopardize the operation.

In addition, a methodology was developed to determine the minimum charge level to which the battery can be brought considering the relationship between the thrust control signal and the battery voltage. This allows knowing the available flight time, and moreover, if we combine this with the estimate of the energy needed to execute a mission, we can

determine the flight time during which the UAV will be able to operate with the expected remaining energy.

The main contributions of this research work are summarized as follows:

1.  We developed a required-energy estimation system capable of adapting to persistent variations in energy consumption based on fuzzy C-means.
2.  We propose a new methodology to determine the aircraft flight time, which to the best of our knowledge, is the first to consider the effect of the battery's state of charge on the control signals.

The rest of the paper is organized as follows. Section 2 presents the works related to the estimation of energy required during a flight and the estimation of flight time. Section 3 presents the proposed methodology divided into energy estimation (Section 3.2), state-of-charge estimation (Section 3.3) and flight-time-margin estimation (Section 3.4). Section 4 shows the application of the proposed methodology in a hexarotor UAV. Finally, Section 5 presents the conclusions and future improvements that can be applied to the proposed methodology.

## 2. Related Work

In the process of estimating the energy required to conduct a specific mission, three different approaches can be distinguished: (i) methods based on aerodynamic models, (ii) methods using regressions, and (iii) those based on intelligent systems. Some of the principal solutions that have been developed in the field of energy estimation are discussed below.

One of the most widely used models is presented in [14]. This model provides a simple way to approximate the required power based on the total weight of the vehicle, its displacement speed, the efficiency in the transfer of energy from the motor to the propeller, and the drag–lift ratio of the vehicle, in addition to a term corresponding to the power consumed by the vehicle's electronics. While this methodology provides an easy way to estimate the consumed energy, it neglects significant factors such as the wind and the air density, which could affect the vehicle's energy consumption. Therefore, this methodology should be used with caution when determining the energy required for a specific mission.

In [15], the authors proposed a power-estimation method wherein the vehicle motion is decomposed into its horizontal and vertical components. For each component, the required power is evaluated considering the acceleration and velocity at which it moves. The aerodynamic effect is considered assuming that the reference surface will have the characteristics of a flat plate. Although the presented model showed favorable performance in numerical simulations, it is complex to find the area affected by the airflow, which will depend on the direction of flight, and the wind speed and direction. In addition, parameters such as propeller tip speed are not available for most UAVs.

A simulation model was presented in [16] where aerodynamic, motor, and battery models are considered. For the estimation of the torque required by each rotor, the blade element moment theory is used, from which the consumed power is determined considering the efficiency of the motor. An equivalent circuit model is used for the battery, considering that the effective capacity is determined using a correction factor as a function of the required power. Finally, the flight time is calculated by dividing the effective battery energy by the required power. This method, despite the positive relation between the measured results and those obtained by a simulation, was not conceived as a method for online energy estimation during a mission.

Regression-based estimation methods, such as the one presented in [17], estimate the required energy based on the vehicle's operation. The authors divided the mission phases into: the idle mode, armed, takeoff, vertical and horizontal flight, and the effect of the payload. For each of these stages, a polynomial regression was performed based on data obtained from experimental tests. Although this method provides an easy way to estimate the energy required to complete a mission, it is not able to adapt to conditions different from those of the flights in which the modeling data were obtained.

The method presented in [18] uses a set of regressors using elastic net regression. The regressors were set up in two stages for ascent, descent, and horizontal movement. The first stage determines the time during which a maneuver will be performed, and the second stage determines the energy required based on the determined time. In the second stage, the required energy for the moments when the vehicle is in hovering flight is added. Although this method showed high accuracy in energy estimation, the method does not include a way to adapt its estimation depending on the mission performance.

The problem of the lack of adaptability to variations in consumption can also be observed in [19], where the authors employed a deep neural network that used as inputs the velocity, acceleration, altitude, wind speed, weight, and surface of the load. Although the proposed method considers multiple variables, which increases its accuracy in energy estimation, it is also unable to adapt to changes not considered in the model's training.

As can be seen in the literature review, data-driven energy-estimation methods, the ones using regression analysis and the ones that employ neural networks, have shown good performance for applications of energy estimation during a mission. However, it is necessary to solve the lack of adaptability of the presented techniques to conditions not considered during their training.

With respect to the flight time, it will depend on the energy required and the available energy. Some of the principal methodologies developed to determine the flight time are discussed below.

In [20], a method is presented to estimate the flight time using regressions and deep learning, considering factors such as known flight time without load, payload, battery capacity, and the onboard computer. However, this method only allows an a priori estimation, since it does not provide a way to update the estimation during the execution of the mission.

In [21,22], the flight time was obtained from the division of the battery capacity by the discharge rate, where it was assumed that the available energy is known. However, there are factors that can modify the amount of usable battery energy that must be considered to provide an accurate estimation of the flight time.

Based on the above methodologies for flight time estimation, it can be observed that it is necessary to have solutions to dynamically adapt the flight time estimations considering that the usable energy may change in situations such as increasing of the payload weight, adverse weather conditions, or system failures.

## 3. Methods

Energy estimation for a mission is a complex problem due to the multiple factors and variables involved in the process. Nevertheless, to estimate the energy requirements is crucial to guaranteeing the feasibility and safety of a mission. Moreover, given the dynamism of the environment in which a multirotor UAV can fly, and the variations to which it may be subject either by lowered energy efficiency due to wear of its components or malfunctioning of one of its parts, it is necessary to continually reevaluate the energy required to complete the mission. In addition, the knowledge of the flight time during which the multirotor UAV can continue operating can be used in decision-making process by an autonomous system or by a human operator.

The architecture of our proposed system is depicted in Figure 1. The system works with a methodology consisting of three parts, which are described as follows:

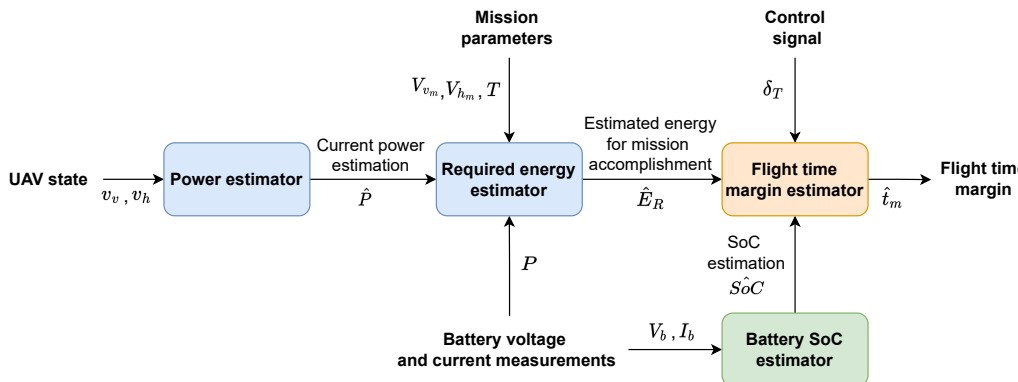

**Figure 1.** Overview of the available energy and flight time estimation process. The system consists of three sections, energy estimation, battery-charge estimation, and flight-time-margin estimation.

(i) The first part of the methodology corresponds to the estimation of the energy required in the mission once the vehicle is in flight, based on the knowledge of the horizontal and vertical velocities at which it will move along the different stages of the mission, and the time during which it will move at that velocity. To do this, we propose the use of a cascaded Takagi–Sugeno fuzzy system, using the C-means algorithm for the premise of the rules, to estimate the power required to move at a given speed. From the knowledge of the power required to move at a given velocity and the time during which it performs such action, it is possible to know the energy required for the mission. It is also proposed to use the PSO algorithm for the optimization of the parameters used in the fuzzy C-means algorithm to find a balance between the execution time of the system, which is affected by the number of clusters, and the accuracy of the system, which is affected by both the number of clusters and the weighting exponent.

(ii) The second part of the methodology consists of determining the state of charge of the battery, for which it is proposed to use an extended Kalman filter based on the equivalent circuit model of the battery, for which it is proposed to use fuzzy modeling to define the relationship between the open-circuit voltage and the state of charge.

Finally, (iii) the third part of the methodology consists of determining the flight-time margin considering the effect of the battery's voltage change during discharge on the thrust control signal. For this stage, it is proposed to use recursive least-squares to determine the relationship between the battery voltage and the thrust control signal in order to determine the minimum voltage at which the vehicle can operate considering a maximum value for the average value of the thrust control signal. From this voltage, we determine the associated battery charge level considering also the constraints given by the operator. Finally, based on the knowledge of the energy required to complete the mission, the battery charge, and the minimum allowable charge level, we calculate the flight time margin.

The methods used in each of the stages are detailed below.

### 3.1. Preliminaries

A brief introduction to Takagi–Sugeno fuzzy systems with premises given by trapezoidal membership functions and using the C-means method is presented, and a way to determine the values of the consequent parameters of the fuzzy rules based on the membership values and output values of a training data set. Additionally, we present the operation of the particle swarm optimization algorithm with a constraint factor on the particle velocity.

#### 3.1.1. Takagi–Sugeno Fuzzy Systems with Fuzzy C-Means

Takagi–Sugeno (T-S) fuzzy systems have been adopted in several applications, since they are able to approximate a function with adequate accuracy in a closed set, maintain

a structure that is easy to interpret thanks to its high transparency, and maintain low computational complexity [23]. These systems are defined by a set of $M$ rules in the form:

$$\textbf{If} \quad x \quad \text{is} \quad A_i, \quad \textbf{Then} \quad y_i = a_i x + b_i \tag{1}$$

where the premise of the rule is formed by the input of the system $x$ and the fuzzy set $A_i$, and the consequent of the rule is defined as $y_i$, $a_i$ and $b_i$ being design parameters.

The output of the T-S fuzzy system is given by

$$y = \frac{\sum_{i=1}^{M} u_i y_i}{\sum_{i=1}^{M} u_i} \tag{2}$$

where $u_i \in [0,1]$ indicates the degree of membership of the input $x$ to each of the fuzzy sets $A_i$.

The form of determining the degree of membership will depend on the way in which the fuzzy set is defined. One of the most common used ways to define the fuzzy set is with a trapezoidal function [24], such as the one shown in Figure 2.

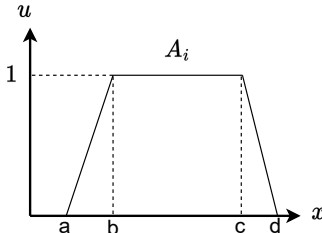

**Figure 2.** Trapezoidal membership function. This function is defined by four points and will have a value of $0 < u_i \le 1$ for $a < x < d$.

In these sets, the level of membership is calculated as:

$$u_i = max\left( min\left( \frac{x-a}{b-a}, 1, \frac{d-x}{d-c} \right), 0 \right) \tag{3}$$

Another way to determine the fuzzy sets is through the fuzzy C-means method, which determines the membership value of $x \in \mathbb{R}^n$ to a fuzzy set, from the closeness to the cluster center, defined by a vector $v \in \mathbb{R}^n$ [25]. For a set of $N$ data, grouped in $M$ fuzzy sets, we must minimize the function

$$J_m = \sum_{k=1}^{N} \sum_{i=1}^{M} (u_{ik})^m \|x_k - v_i\|^2 \tag{4}$$

where $m > 1$ is the weighting exponent, which is a design parameter that modifies the performance of the fuzzy C-means method.

The membership value used in (4) is given by

$$u_i = \frac{1}{\sum_{j=1}^{M} \left( \frac{\|x_i - v_i\|}{\|x_i - v_j\|} \right)^{\frac{2}{m-1}}} \tag{5}$$

where the optimal values of the centers $v$ for a given number of clusters $M$ and exponent $m$ are obtained through the iterative process presented in [25].

### 3.1.2. Computation of Parameters for the Consequents in T-S Systems

Consider a set of training input data $X \in \mathbb{R}^{N \times n}$, for which the membership value $U_i \in \mathbb{R}^N$ corresponding to each of the $M$ fuzzy sets $A_i$ is known, and the expected output of

the fuzzy system is $Y \in \mathbb{R}^N$. It is possible to employ the least-squares method to determine the design parameters $a_i$ and $b_i$ of each of the fuzzy rules as follows [26].

Let us define the extended matrix $X_e$ as:

$$X_e = \begin{bmatrix} X & \mathbf{1} \end{bmatrix} \tag{6}$$

where $\mathbf{1} \in \mathbb{R}^N$ is a vector of ones. Then, using the matrix $X_e$ and the membership values $U_i$, the following matrix is formed:

$$X' = \begin{bmatrix} \text{diag}(U_1)X_e & \cdots & \text{diag}(U_M)X_e \end{bmatrix} \tag{7}$$

Finally, with a global approach using the $X'$ matrix, the vector of parameters for the consequents will be calculated as follows.

$$\theta = [(X')^T X']^{-1} (X')^T Y \tag{8}$$

where the values obtained in the parameter vector have the following structure:

$$\theta = \begin{bmatrix} a_1 & b_1 & \cdots & a_M & b_M \end{bmatrix} \tag{9}$$

which correspond to the design parameters required in the consequent of each of the rules.

### 3.1.3. Particle Swarm Optimization

The particle swarm optimization (PSO) method is a bio-inspired optimization technique that was presented in [27]. In this method, a set of particles $p_i \in \mathbb{R}^n$ is proposed, which are moved through a search space to find the minimum (or maximum) of a function $J(p_i)$. The displacement is performed based on the best solutions found by each of the particles, which are initialized with random values within a search space.

In [28], a variant of the original PSO method was developed to improve the convergence capabilities. In this variant, the velocity of each particle $v_i$ at instant $k$ is calculated as:

$$v_i[k] = \chi(v_i[k-1] + c_1\text{rand}()(p_{b_i} - p_i[k-1]) + c_2\text{rand}()(p_{b_g} - p_i[k-1])) \tag{10}$$

where $\chi$ is a velocity constraint factor, $c_1$ and $c_2$ are constants that weigh the effect of the cognitive and social components, $p_{b_i}$ is the value of the particle that generated the minimum of that particle, and $p_{b_g}$ is the value of the particle that generated the global minimum among the particles. The factor $\chi$ is calculated as a function of $c_1$ and $c_2$ as:

$$\chi = \frac{2}{|2 - \phi - \sqrt{\phi^2 - 4\phi}|} \quad , \quad \phi = c_1 + c_2 \quad , \quad \phi > 4 \tag{11}$$

Given the velocity of each particle, the position of each particle is updated as:

$$p_i[k] = v_i[k] + p_i[k-1] \tag{12}$$

In order to find the minimum of $J(p_i)$, an iterative process of velocity calculation and position update of each particle is performed, in which the values of $p_{b_i}$ and $p_{b_g}$ are acquired. The updating of $p_{b_g}$ can be performed synchronously—i.e., this value is updated when the cost function for all particles has been evaluated; or it can be asynchronous, where the value of $p_{b_g}$ is updated each time a new global minimum is encountered, regardless of whether the iteration has not been completed. The performance of the synchronous and asynchronous methods was evaluated in [29], where it was shown that the asynchronous method improves the convergence speed of the method.

### 3.2. Estimation of the Required Energy in Flight

As discussed above, the energy consumption of a multirotor UAV depends on a wide variety of factors, and since the fuzzy C-means technique performs well for systems up to about 20 dimensions [30], it is used in the development of the energy estimation system.

#### 3.2.1. System for the Estimation of the Required Energy

To estimate the required energy during a mission, it is assumed that during the development of a mission with automatic navigation, the vehicle follows a series of waypoints, which consist of coordinates related to information of the flight altitude, speed, and hover time. To cover the points defined in a desired trajectory, the vehicle will move following velocity profiles, which can be decomposed into their vertical and horizontal components. Based on the behavior of the navigation algorithm used, it is possible to anticipate the velocity profile that will be present along the trajectory, and therefore, the proposed energy estimation system will use this information. It consists of two cascaded subsystems which use fuzzy clustering, as shown in Figure 3.

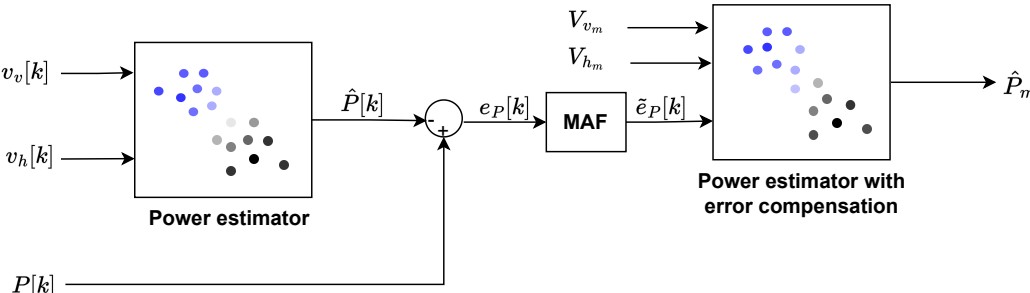

**Figure 3.** Proposed system for the estimation of the required power during a flight. It is composed of two cascaded subsystems based on fuzzy clustering. The first subsystem must determine the required power for the current system velocity, and the second subsystem determines the required power for subsequent mission stages based on the expected velocities and the determined power-estimation error.

The first subsystem evaluates the current state of the system, having as input the vertical velocity $v_v$, positive in the downward direction; and the horizontal velocity, $v_h \geq 0$. The output corresponds to the power estimation ($\hat{P}$) for the present state of the vehicle. Such power estimation is performed continuously as the flight develops. The value of $\hat{P}$ is determined from a set of $M_1$ fuzzy rules of the form

$$\textbf{If} \quad [v_v \quad v_h] \quad \text{is} \quad A_i, \quad \textbf{Then} \quad y_i = a_{1i}v_v + a_{2i}v_v + b_i \tag{13}$$

From the defined rules, the value of $\hat{P} = y$ is calculated using Equation (2). The output of the first subsystem is used in combination with the power measurement $P$, obtained from the UAV sensors, to calculate the power-estimation error as:

$$e_p = P - \hat{P} \tag{14}$$

The estimation error $e_p$ will be smoothed by a moving average filter (MAF) given by:

$$\tilde{e}_p = \frac{1}{j} \sum_{i=k-j}^{k} e_p \tag{15}$$

where $j$ corresponds to the number of data samples used in the filter.

The second subsystem utilizes as input the sets of vertical velocities $V_{v_m} = \{v_{v_1}, \cdots, v_{v_s}\}$ and horizontal velocities $V_{h_m} = \{v_{h_1}, \cdots, v_{h_s}\}$ of the $s$ segments that constitute the expected

velocity profile for the rest of the mission, and $\tilde{e}_p$. This subsystem is formed by a set of $M_2$ fuzzy rules of the form:

$$\text{\textbf{If}} \quad \begin{bmatrix} v_{v_i} & v_{h_i} & e_p \end{bmatrix} \quad \text{is} \quad A_i, \quad \text{\textbf{Then}} \quad y_i = a_{1i}v_v + a_{2i}v_h + a_{3i}e_p + b_i \tag{16}$$

The output of this subsystem, given by Equation (2), corresponds to the required power to fly at the velocities defined in the different stages of the mission $\hat{P}_m = \{\hat{P}_{m_1}, \cdots, \hat{P}_{m_s}\}$.

Finally, the estimation of the energy required to complete a mission is given by

$$\hat{E}_R = \sum_{i=1}^{s} \hat{P}_{m_i} T_i \tag{17}$$

where $T_i$ corresponds to the period during the multirotor UAV moves at velocities $v_{v_i}$ and $v_{h_i}$.

3.2.2. Computation of the Parameters of the Power-Estimation System

The performance of the proposed subsystems for power estimation depends on the quality of the training set, the number of clusters, and the chosen fuzzy exponent ($m$). There is also a trade-off between the number of clusters and the accuracy of the power estimation, since a larger number of clusters can lead to more accurate results, but, in a resource-constrained application, such as in embedded systems, it is necessary to employ algorithms with low computational cost that can be executed in a short period of time. In order to find a balance between power-estimation accuracy and speed of execution, we propose the use of the PSO method presented in Section 3.1.3.

In this sense, the structure of the particles to be used during the optimization process is defined as follows:

$$p_i = \begin{bmatrix} M'_i & m_i \end{bmatrix} \tag{18}$$

where $M'_i$ is an auxiliary variable that allows one to define the number of clusters, and $m_i$ is the weighting exponent associated with the particle. In order to carry out the PSO algorithm, the following cost function is proposed:

$$J = w_1 \frac{1}{N} \sum_{i=1}^{N} |e_p| + w_2 \lfloor M'_i \rceil \tag{19}$$

where $N$ is the number of samples in the validation set, and $w_1$ and $w_2$ are constants that represent the trade-off between system accuracy and execution speed. The function $\lfloor \cdot \rceil$ indicates the value rounded to the nearest integer. Although the above function does not explicitly include the $m_i$ parameter of the particle, it is reflected through $e_p$.

For the first subsystem, the training data set will be composed of the vertical velocities $V_v = \{v_{v_1}, \cdots, v_{v_N}\}$, horizontal velocities $V_h = \{v_{h_1}, \cdots, v_{h_N}\}$, and measured power $P = \{P_1, \cdots, P_N\}$. To apply the PSO method to the subsystem, the clustering process [25] is performed with $x = \begin{bmatrix} v_{v_i} & v_{h_i} \end{bmatrix}$ and $y = P_i$. The number of clusters is given by $M_1 = \lfloor M'_i \rceil$ and the weighting exponent $m_1 = m_i$.

Using the obtained cluster centers $v_i$, the membership values are calculated with Equation (5) for the training set, obtaining a set $U_i = \{u_{i1}, \cdots, u_{iN}\}$ for each cluster. The obtained values are used in the least-squares method presented in Section 3.1.2 with $X = \begin{bmatrix} V_v & V_h \end{bmatrix}$ and $Y = P$, to obtain the parameters of the consequent of the rule shown in Equation (13). Once the parameters of the $M_1$ rules have been determined, using Equation (2), the values of $\hat{P} = \{\hat{P}_1, \cdots, \hat{P}_i\}$ are calculated for the validation set. Finally, the function (19) is calculated with the values of $e_p$, as given in (14) between the validation set measurements and the power estimations.

The process described above is conducted for each particle until the defined maximum number of iterations is reached. The values of $M_1$ y $m_1$ will correspond to the last value obtained for $p_{b_g}$.

For the training of the second subsystem, we calculate the set $\tilde{E}_p = \{\tilde{e}_{p_1}, \cdots, \tilde{e}_{p_N}\}$ using Equation (15) for the values of $e_p$ obtained from the power-estimation process of the training data set used in the first subsystem.

The training of the second subsystem is performed using the training data set of the first subsystem, extended with $E_p$, $x = [v_{v_i} \quad v_{h_i} \quad e_{p_i}]$ and $y = P_i$. The training process is performed using the procedure indicated for subsystem one to obtain $M_2$ and $m_2$, and the parameters in the consequent of each rule (16).

### 3.3. Estimation of the Battery's State of Charge

This section presents the mathematical model of the equivalent circuit used for batteries using a single time constant, proposing the use of fuzzy modeling for the relationship between the battery's open-circuit voltage and the state of charge. After that, the estimation of the state of charge using an extended Kalman filter is addressed.

#### 3.3.1. Equivalent Circuit Model for Batteries

An electric battery is an element that stores energy in chemical form, which can be released in a controlled way. A model that has been widely used to determine its behavior is the equivalent circuit model [31]. Figure 4 presents the model of a single time constant, which allows studying its behavior with a suitable degree of accuracy when the objective is the estimation of the state of charge (*SoC*) for practical applications [32].

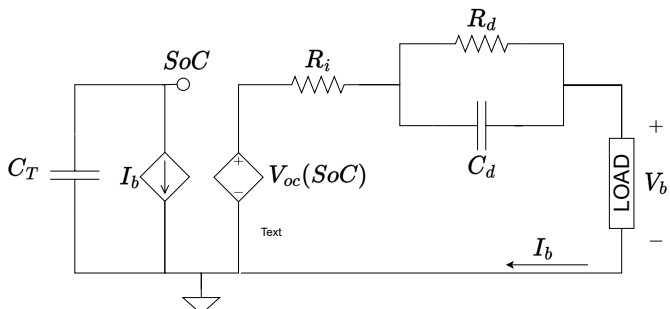

**Figure 4.** Equivalent circuit model with a single time constant for batteries.

On the left-hand side, a capacitor $C_T$ models the battery charge capacity, and a current source whose flow is equal to that flowing through the load at the battery terminals $I_b$. The capacitance of $C_T$ is given by:

$$C_T = 3600 Q_B \eta \tag{20}$$

where $Q_B$ is the capacity of the battery expressed in *Ah* and $\eta$ is a factor that will depend on the temperature and health of the battery. The voltage at $C_T$, whose value is between zero and one, represents the *SoC* of the battery and is calculated by:

$$SoC(t) = SoC(t_0) - \frac{1}{C_T} \int_{t_0}^{t} I_b(\tau) d\tau \tag{21}$$

On the right-hand side, a voltage source models the open-circuit voltage $V_{oc}$ as a function of the *SoC*. $R_i$ is the internal resistance, and the pair $R_d C_d$ represents the transient behavior of the battery. The dynamics of the voltage $\dot{V}_d$ on $R_d C_d$, and the battery terminal voltage $V_b$, are given by:

$$\dot{V}_d = \frac{I_b}{C_d} - \frac{V_d}{R_d C_d} \tag{22}$$

$$V_b = V_{oc}(SoC) - I_b R_i - V_d \tag{23}$$

The function between the $V_{oc}$ and the $SoC$ has been approximated from a set of linear functions or polynomial functions [33,34]. In the present work, it is proposed to approximate such a function with a Takagi–Sugeno fuzzy model with rules of the form given in Equation (1), as shown in Figure 5. It can be observed that the function is segmented using trapezoidal fuzzy sets. This approximation allows having the simplicity given by a set of linear functions while maintaining the smoothness of the transitions between regions of the curve, as can be observed in polynomial approximations.

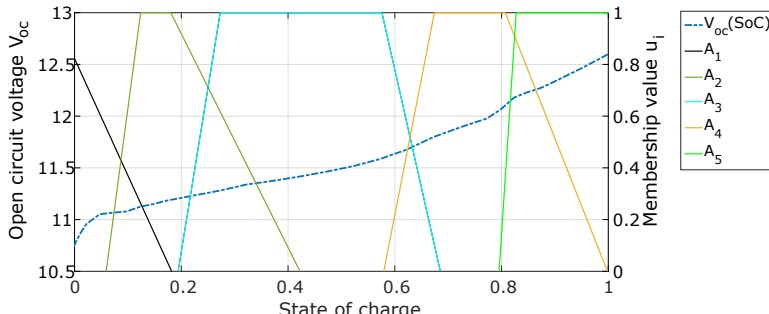

**Figure 5.** Modeling the relationship between the $V_{oc}$ and $SoC$ using a set of trapezoidal fuzzy sets.

Each membership function of the set of rules defining the functions $V_{oc}(SoC)$ or $SoC(V_{oc})$ is given by four points as seen in Figure 2, whose membership value will be given by Equation (3), and the output of the model is given by Equation (2).

### 3.3.2. Estimation of the *SoC*

A widely used methodology for the estimation of the *SoC* of the battery is the use of the Kalman filter, which, unlike the Coulomb count, allows compensating the differences between the estimated initial *SoC* and the real one; moreover, it presents lower vulnerability to noise in the current measurements [35].

To implement the extended Kalman filter (EKF) [36], Equations (21) and (22) are discretized using Euler's method and grouped as follows:

$$\underbrace{\begin{bmatrix} SoC[k] \\ V_d[k] \end{bmatrix}}_{x[k]} = \underbrace{\begin{bmatrix} 1 & 0 \\ 0 & \left(1 - \frac{\Delta t}{R_d C_d}\right) \end{bmatrix}}_{A} \underbrace{\begin{bmatrix} SoC[k-1] \\ V_d[k-1] \end{bmatrix}}_{x[k-1]} + \underbrace{\begin{bmatrix} -\frac{\Delta t}{C_T} \\ \frac{\Delta t}{C_d} \end{bmatrix}}_{B} \underbrace{I_b[k-1]}_{u[k-1]} \tag{24}$$

where $\Delta t$ is the sampling time interval. In Equation (24), the terms of the battery state-space model $x$, $u$, $A$, and $B$ are obtained. In addition, from the linearization of Equation (23), we obtain:

$$\underbrace{V_b[k]}_{y[k]} = \underbrace{\begin{bmatrix} \frac{\partial V_{oc}}{\partial SoC}[k] & -1 \end{bmatrix}}_{C[k]} \underbrace{\begin{bmatrix} SoC[k] \\ Vd[k] \end{bmatrix}}_{x[k]} + V_{oc}(SoC[k]) - \frac{\partial V_{oc}}{\partial SoC}[k] SoC[k] - \underbrace{R_i}_{D} \underbrace{I_b[k]}_{u[k]} \tag{25}$$

Using the terms obtained in (24) and (25), the *SoC* of the battery is calculated by employing the EKF given in Algorithm 1, where $P$ is the covariance error matrix, $K$ is the estimator gain, $Q$ is the process noise covariance, and $R$ is the measurement noise covariance. The value of $\frac{\partial V_{oc}}{\partial SoC}$ can be obtained from the fuzzy model for the function $V_{oc}(SoC)$ using the partial derivative of its consequent.

### 3.4. Estimation of Flight Time Margin

This section presents an analysis of the relationship between the battery charge and the thrust control signal, from which a methodology is developed to determine the minimum charge level at which it is possible to operate the multirotor UAV and the time for which it can operate after the end of its mission with the remaining energy.

---

**Algorithm 1** Estimation of the *SoC*.

---

**Require:** $P[k-1] = P_0$, $\hat{x}[k-1] = x_0$, $Q$, $R$

1: **loop**

2:      Perform measurement of $u[k] = I_b[k-1]$ and $y = V_b[k]$

3:      Obtain the value of $\frac{\partial V_{oc}}{\partial SoC}$

4:      Predict the estimated state

$$\hat{x}^-[k] = A\hat{x}[k-1] + Bu[k-1] \tag{26}$$

5:      Prediction of the covariance error

$$\hat{P}^-[k] = AP[k-1]A^T + BQB^T \tag{27}$$

6:      Calculate the Kalman gain

$$K = \hat{P}^-[k]C^T(C\hat{P}^-[k]C[k]^T + DRD^T)^{-1} \tag{28}$$

7:      Update the estimated state estimate with the measurement of $y[k]$

$$\hat{x}[k] = \hat{x}^-[k-1] + K(y[k] - y(\hat{x}^-[k], u[k-1])) \tag{29}$$

8:      Update covariance error

$$P[k+1] = (I - K[k]C)P^-[k] \tag{30}$$

9: **end loop**

---

### 3.4.1. Relationship between Control Signals and the *SoC*

In the controller field of multirotor UAVs, a widely used architecture is the one that uses a set of cascaded controllers, as shown in Figure 6 [37,38]. This architecture begins with a position controller for a reference $\xi_d$, from which an attitude reference $\eta_d$ is obtained, along with a thrust control signal $\delta_T$. From the previous reference, an attitude controller generates an angular velocity reference $\omega_d$. The angular velocity reference is sent to an angular velocity controller that will generate a set of control signals for the angular velocity $\delta_p$, $\delta_q$, and $\delta_r$. The four control signals obtained are used by a control allocation system, also known as mixer, which will generate a set of control signals $U$ for each of the rotors of the multicopter.

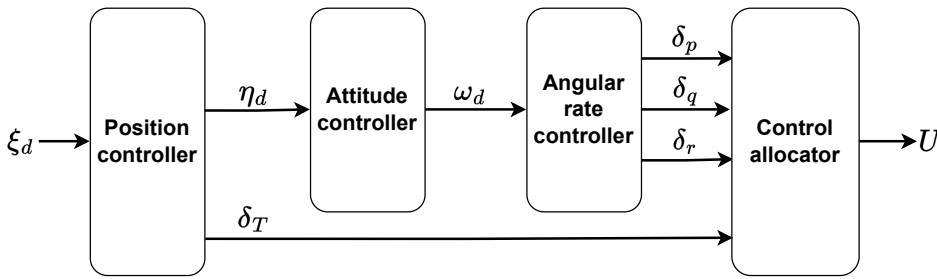

**Figure 6.** Control architecture employed in widely used autopilots such as PX4 or Arducopter.

For the operation of a multirotor UAV at a constant altitude, the thrust control signal is of a magnitude to provide the force required to compensate for the weight of the multirotor UAV, for which there is a control signal for each of the rotors. The angular velocity control signals will generate variations in the control signals of each rotor around the point required to generate the thrust force. The force generated by each rotor $f_r$ is calculated as:

$$f_r = k_f(k_\omega V_m)^2 \tag{31}$$

where $k_f$ is the propeller force constant, $k_\omega$ is the motor angular velocity constant, and $V_m$ is the average voltage applied to the motor.

As can be seen in Equation (31), for a given rotor, the force generated will depend on the average voltage applied. In turn, the average voltage applied to the rotor will be proportional to the duty cycle of the applied signal and is computed as [39]:

$$V_m = V_b u_d \tag{32}$$

where $u_d$ is the duty cycle with $0 \le u_d \le 1$.

As can be seen in Equation (32), there is a linear relationship between the duty cycle and the battery voltage. Therefore, for a given flight condition, the duty cycle will increase as the battery voltage decreases. For a normalized thrust control signal ($\delta_T$) we have the behavior shown in Figure 7, where it is observed that the $\delta_T$ control signal increases linearly as the voltage ($V_b$) decreases.

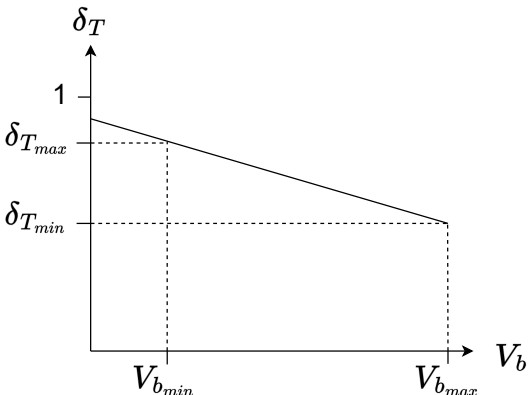

**Figure 7.** Relationship between battery voltage and thrust control signal. For a constant flight condition, a linear relationship will be maintained between both values.

The $\delta_T$ value is minimal when the battery presents its maximum voltage, and should not exceed a maximum level, which allows one to maintain a safe operating margin for variations due to angular velocity controls.

Since the applied voltage will depend on the *SoC* of the battery, as seen in Figure 8, the minimum allowable battery voltage will determine the minimum *SoC* at which it is safe to operate.

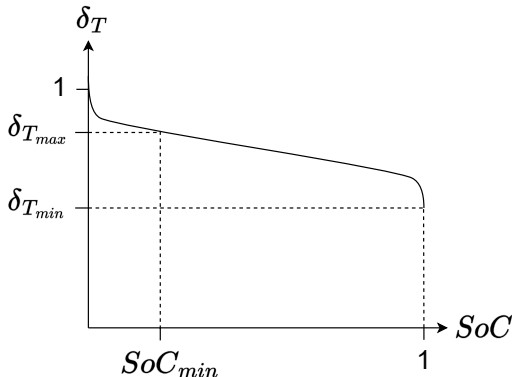

**Figure 8.** Relationship between *SoC* and thrust control signal. For a constant flight condition, the required $\delta_T$ increases depending on the battery discharge curve.

### 3.4.2. Estimation of the Flight Time Margin

The remaining flight time is approximated from the average power consumed $P_m$ and the usable energy in the battery. To determine the usable energy in the battery, the minimum

charge level at which it is safe to operate is determined based on the expected performance of the thrust control signal.

Using a recursive least squares (RLS) [40] process, the following relationship is determined:

$$V_{b_\delta} = \alpha_1 \delta_T + \alpha_2 \tag{33}$$

where $V_{b_\delta}$ is the expected battery voltage for a given $\delta_T$, and $\alpha_1$ and $\alpha_2$ are parameters obtained from the RSL process.

Then, using the maximum admissible value of the thrust control signal $\delta_{T_{max}}$, the minimum admissible voltage at the battery terminals $V_{b_{min}}$ is calculated as:

$$V_{b_{min}} = \max\{V_{b_s}, V_{b_\delta}(\delta_{T_{max}})\} \tag{34}$$

where $V_{b_s}$ is the minimum voltage at which the battery can be safely operated.

Based on the equivalent circuit model (Figure 4), it can be seen that in the worst-case scenario, the open-circuit voltage when the terminal voltage is $V_{b_{min}}$, for a value of $P_m$, is given by:

$$V_{oc_{min}} = V_{b_{min}} + \frac{P_m}{V_{b_{min}}}(R_i + R_d) \tag{35}$$

Therefore, the minimum charge level at which it is possible to operate the multirotor UAV under the constraints of $\delta_{T_{max}}$ is given by:

$$\hat{SoC}_{min} = \max\{SoC_s, SoC(V_{oc_{min}})\} \tag{36}$$

where $SoC_s \geq 0$ is the minimum charge at which it is desired to operate the battery.

For a defined mission whose required energy is given by $\hat{E}_R$ when the battery has a charge $SoC_0$, the estimated charge at the end of the mission $\hat{SoC}_f$ is calculated as:

$$\hat{SoC}_f = SoC_0 - \frac{\hat{E}_R}{E_T} \tag{37}$$

where $E_T$ is the energy stored in the battery when the battery is fully charged and is calculated by:

$$E_T = Q_B V_{b_{nom}} \tag{38}$$

where $V_{b_{nom}}$ is the nominal voltage of the battery.

A mission may be considered as an achievable mission if $\hat{SoC}_f \geq SoC_{min}$. If the mission is achievable, then the flight time margin $\hat{t}_m$ is estimated as:

$$\hat{t}_m = \frac{(\hat{SoC}_f - SoC_{min})E_T}{P_m} \tag{39}$$

In the case of a manually controlled flight, this method may be used to estimate the remaining flight time using $\hat{E}_R = 0$.

## 4. Physical Experiments and Results

In order to evaluate the performance of the proposed methods, a series of physical experiments were carried out using a hexarotor UAV as the case of study. The experiments consisted of (i) performing discharge tests of the battery used on the hexarotor UAV in order to obtain the parameters of its equivalent circuit and the relationship between the open-circuit voltage and the state of charge; (ii) executing a series of flights to obtain information for the training process of the power-estimation system, including the optimization of its parameters; and finally, (iii) evaluating the performance of the proposed methods through a validation flight.

The hexarotor used to conduct the experiments, shown in Figure 9, has the specifications given in Table 1.

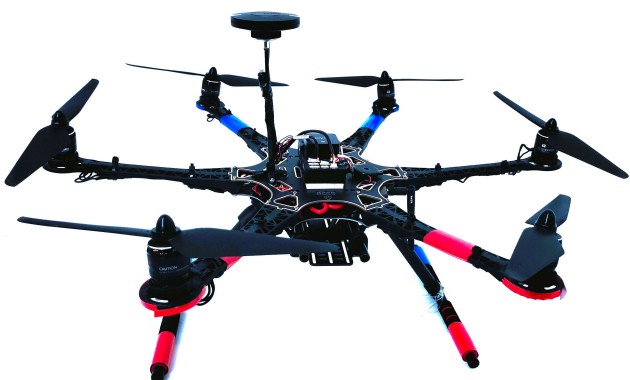

**Figure 9.** Hexarotor UAV used as the case study for the estimation of the required energy.

**Table 1.** Hexarotor specifications.

| Parameter | Values |
| --- | --- |
| Flight controller | Pixhawk 2.1 Cube Black |
| Motor | T-motor Air 2213 920KV |
| ESC | T-motor Air 20A |
| Propeller | T-motor 9.45 × 4.5 in |
| Power module | Mauch HS-050-LV |
| Battery | Turnigy graphene 4Ah 3S 45C |
| Weight | 1.8 kg |
| Dimensions | 55 cm × 55 cm × 23 cm |

The power module uses an Allegro ACS758LCB-050U Hall-effect linear current sensor with a maximum capacity of 50A that is able to provide current measurements with an accuracy of 2%. The development of the aforementioned experiments is detailed below.

### 4.1. Experiment 1: Battery Characterization

The battery used in this experiment was characterized at room temperature using the electronic load model KP-184 shown in Figure 10. The characterization process consisted of performing a series of discharges at constant current. After each discharge, a relaxation period of 15 min was maintained since, after this time, each cell was meant to be within 3 mV of the $V_{oc}$ [41]. This process was performed until an amount of energy equivalent to the nominal capacity of the battery was reached. The discharge process performed to characterize the battery is shown in Figure 11.

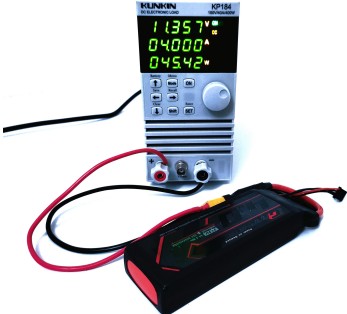

**Figure 10.** KP-184 programmable electronic load used to characterize the battery employed in the experimental platform.

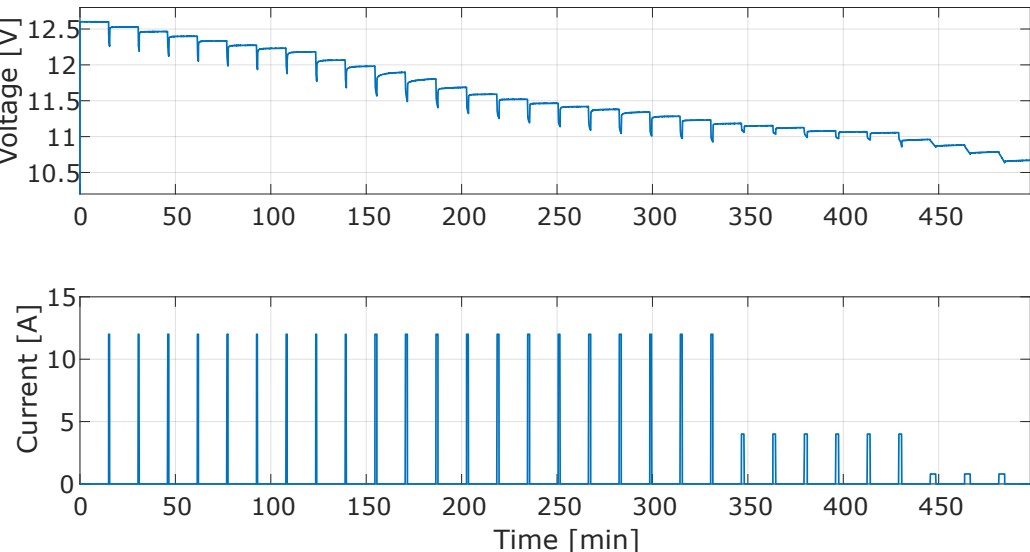

**Figure 11.** Voltage and current graphs obtained from the battery characterization process.

The information obtained from the battery discharge test was analyzed with the process presented in [42], which performed an optimization process using nonlinear least squares to determine the values of the equivalent circuit elements shown in Table 2.

**Table 2.** Equivalent circuit parameters of the used battery.

| Parameter | Value |
|:---:|:---:|
| $R_i$ | 41.6 mΩ |
| $R_d$ | 9.6 mΩ |
| $C_d$ | 1016 F |
| $C_T$ | 14,400 F |

Based on the obtained open-circuit voltage values, two T-S fuzzy models with five rules were generated using trapezoidal membership functions. The first model represents the function $V_{oc}(SoC)$ using the parameters shown in Table 3. The second model represents the inverse function $SoC(V_{oc})$ using the parameters shown in Table 4.

**Table 3.** Parameters of the membership functions and consequents of the rules used for the model that determines the $V_{oc}$ for a given $SoC$. The points correspond to trapezoidal membership functions, as shown in Figure 2.

| | Membership Function Points | | | | |
|:---:|:---:|:---:|:---:|:---:|:---:|
| Rule | a | b | c | d | Consequent |
| 1 | −0.2073 | −0.1045 | −0.2485 | 0.1568 | $V_{oc} = 14.14 SoC + 10.76$ |
| 2 | 0.0033 | 0.0887 | 0.2246 | 0.4007 | $V_{oc} = 2.565 SoC + 10.71$ |
| 3 | 0.1731 | 0.3055 | 0.5641 | 0.6686 | $V_{oc} = 1.325 SoC + 10.83$ |
| 4 | 0.5565 | 0.6668 | 0.7931 | 1 | $V_{oc} = 1.827 SoC + 10.57$ |
| 5 | 0.7793 | 0.8354 | 1.077 | 1.18 | $V_{oc} = 1.912 SoC + 10.69$ |

**Table 4.** Parameters of the membership functions and consequents of the rules used for the model that determines the *SoC* for a given $V_{oc}$. The points correspond to trapezoidal membership functions, as shown in Figure 2.

| | Membership Function Points | | | | |
|---|---|---|---|---|---|
| **Regla** | **a** | **b** | **c** | **d** | **Consecuente** |
| 1 | 9.948 | 10.17 | 10.53 | 10.77 | $SoC = 0.1089V_{oc} - 1.155$ |
| 2 | 10.46 | 10.75 | 10.92 | 11.32 | $SoC = 0.1716V_{oc} - 1.1852$ |
| 3 | 11.05 | 11.07 | 11.59 | 11.78 | $SoC = 0.8967V_{oc} - 9.82$ |
| 4 | 11.57 | 11.83 | 12.21 | 12.43 | $SoC = 0.4059V_{oc} - 4.109$ |
| 5 | 12.23 | 12.43 | 12.77 | 12.99 | $SoC = 0.3735V_{oc} - 3.706$ |

*4.2. Experiment 2: Training of the Energy Estimation System*

To conduct the training of the energy estimation system, we used data from a series of flights, performing various patterns of horizontal and vertical movements outdoors. The flights were performed at an altitude of 2245 m in diverse weather conditions, including winds between 3 and 8 km/h, with gusts of up to 22 km/h. The goal of this experiment was that the energy estimation system would provide an energy estimate as an average of the energy requirements of the different conditions in which it can operate.

The optimization process for obtaining the parameters was performed as indicated in Section 3.2.2, involving a series of 100 iterations for each subsystem. According to the results presented in [29], a set of 30 particles was used for each subsystem. Considering that the values of the exponent *m* must be greater than one, and its upper limit for practical applications is given in [43], the search space is defined by $1.01 \leq m_i \leq 3.5$. Similarly, the search space for the number of clusters was defined to be $2 \leq M'_i \leq 20$ based on the criterion for the maximum number of clusters $M_{max} \leq 2 \ln N$ presented in [44].

Figure 12 presents the value of the cost function (19) evaluated at the value of $p_{b_g}$ for each of the subsystems. Table 5 presents the parameters obtained for the implementation of the first subsystem, and Table 6 shows the parameters obtained for the implementation of the second subsystem.

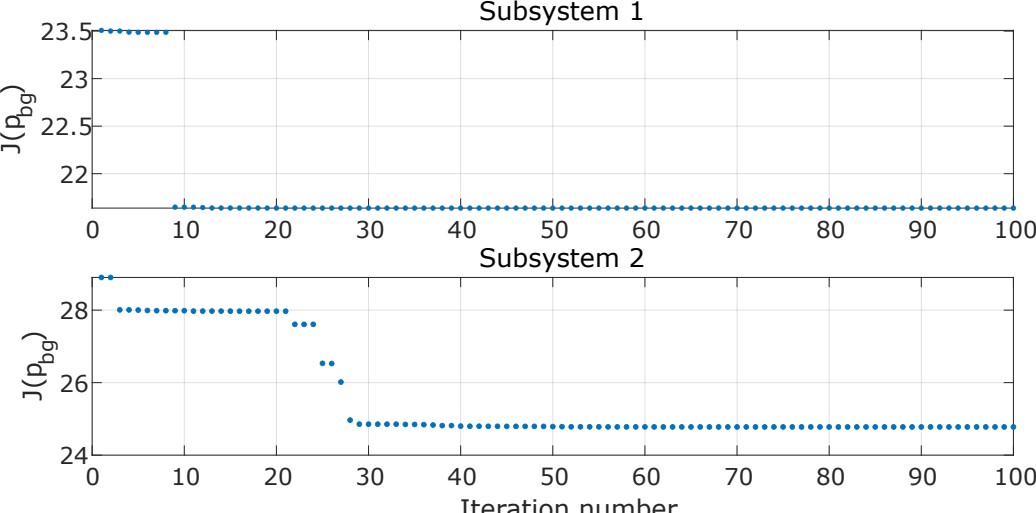

**Figure 12.** Value of the cost function used in the optimization of the power-estimation system, evaluated according to the value of the particle with the best cost of each subsystem.

**Table 5.** Parameters of the first power-estimation subsystem.

| | Number of Clusters $M_1$ | | | Weighting Exponent $m_1$ | | |
|---|---|---|---|---|---|---|
| | 2 | | | 1.4628 | | |
| | **Cluster Centers** | | **Consequent Parameters** | | | |
| **Cluster** | $v_v$ | $v_h$ | $a_{1i}$ | $a_{2i}$ | | $b_i$ |
| 1 | 0.0069 | 4.0424 | −19.011 | −3.5064 | | 262.2819 |
| 2 | −0.0022 | 1.2231 | −17.4258 | −1.5127 | | 258.4393 |

**Table 6.** Parameters of the second power-estimation subsystem.

| | Number of Clusters $M_2$ | | | Weighting Exponent $m_2$ | | | |
|---|---|---|---|---|---|---|---|
| | 2 | | | 1.0338 | | | |
| | **Cluster Centers** | | | **Consequent Parameters** | | | |
| **Cluster** | $v_v$ | $v_h$ | $e_p$ | $a_{1i}$ | $a_{2i}$ | $a_{3i}$ | $b_i$ |
| 1 | −0.0014 | 2.0883 | −12.6536 | −15.6601 | −3.4644 | 0.9661 | 260.9404 |
| 2 | −0.0072 | 1.4581 | 50.9937 | −16.694 | −3.3685 | 0.684 | 277.0185 |

*4.3. Experiment 3: System Evaluation*

To validate the performance of the energy estimation system, we conducted a validation flight. The mission profile consisted of a series of climb and descent maneuvers while performing increments in translation speed, as shown in Figure 13. The flight was executed by using the PX4 system for the vehicle control and navigation.

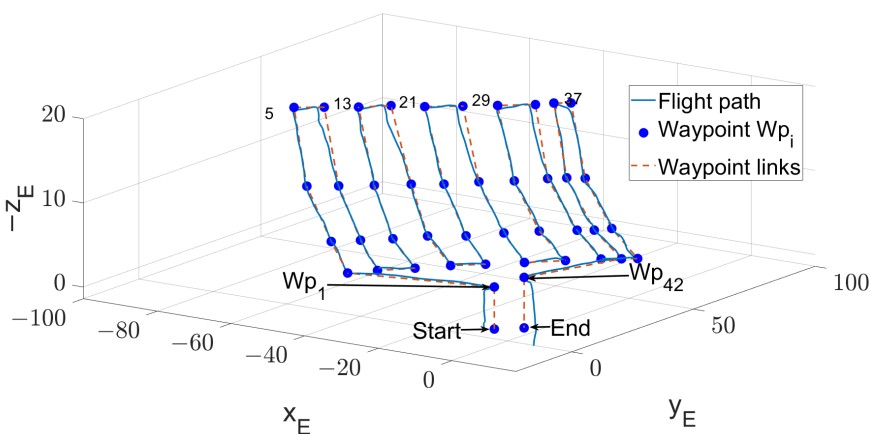

**Figure 13.** Mission profile conducted by the hexarotor UAV to validate the performance of the developed energy estimation system. The blue dots represent the waypoints that defined the multirotor UAV mission during the flight. The dotted orange line is the trajectory linking 42 waypoints, and the solid blue line corresponds to the multirotor UAV trajectory.

The energy required to complete the mission was estimated from each of the waypoints that defined the trajectory. Figure 14 shows the comparison between the required energy prediction $\hat{E}_R$ and the ones measured by the Mauch HS-05-LV sensor $E_R$. Calculating the required energy estimation error $e_{E_R} = E_R - \hat{E}_R$, and dividing this by the average measured power value $P_m$, produces the graph shown in Figure 15, which is interpreted in terms of flight time.

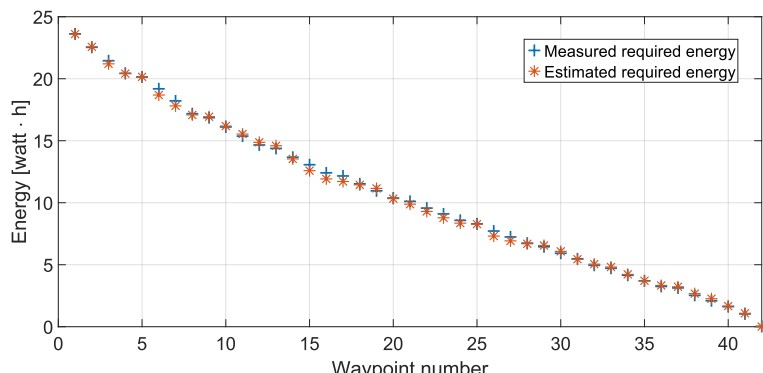

**Figure 14.** Comparison between the measurement of the energy required to complete the mission from a waypoint and the estimation obtained with the developed system.

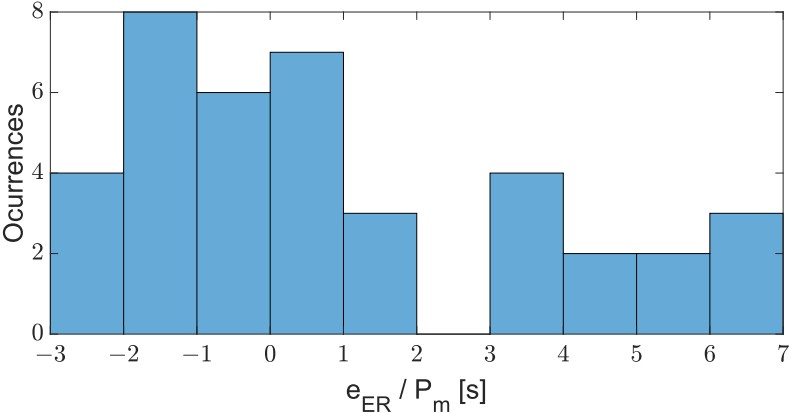

**Figure 15.** Number of occurrences for the values obtained from dividing the energy estimation error by the average required power in the mission. This value can be interpreted in terms of flight time.

From the comparison between the required power estimation and the power use measured by the system, the maximum estimation error was found to correspond to the power required to fly for 7 s at the average power.

As part of the flight time margin estimation process, the RLS method was applied to obtain the relationship given in Equation (33) to determine the expected voltage at the maximum allowable $\delta_T$. In Figure 16, the comparison between the recorded $\delta_T$ and the function $\delta_T(V_b)$ derived from Equation (33) can be observed. As can be seen, the function obtained by the RLS process averages the behavior of $\delta_T$, so it is possible to obtain the $V_b$ for a given average $\delta_T$.

Figure 17 shows the obtained values related to the horizontal and vertical velocities, the attitude, and the estimations of the $S\hat{o}C$, $S\hat{o}C_{min}$, and $\hat{t}_m$ from the validation flight considering a $V_{b_s} = 9.6v$, $SoC_s = 10\%$ and $\delta_{T_{max}} = 0.78$. It is possible to see four intervals where the estimation of the $S\hat{o}C_{min}$ exceeds the established $SoC_s$. These intervals correspond to the instants after changes in the vehicle attitude in the roll, pitch, and yaw angles simultaneously, which increase the control signal $\delta_T$, and consequently, it is estimated that the $\delta_{T_{max}}$ will be reached at a higher $V_b$, so the $S\hat{o}C_{min}$ is higher. The value of $\hat{t}_m$ was computed by considering an $\hat{E}_R = 0$, in which it is possible to observe intervals where the flight time estimation decreases due to the increase in $S\hat{o}C_{min}$.

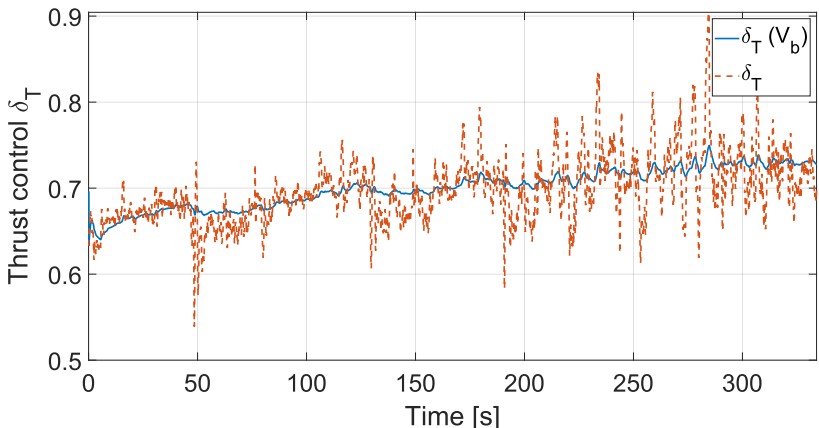

**Figure 16.** Comparison between the control signal $\delta_T$ recorded during the validation flight and the one obtained from the inverse function of Equation (33).

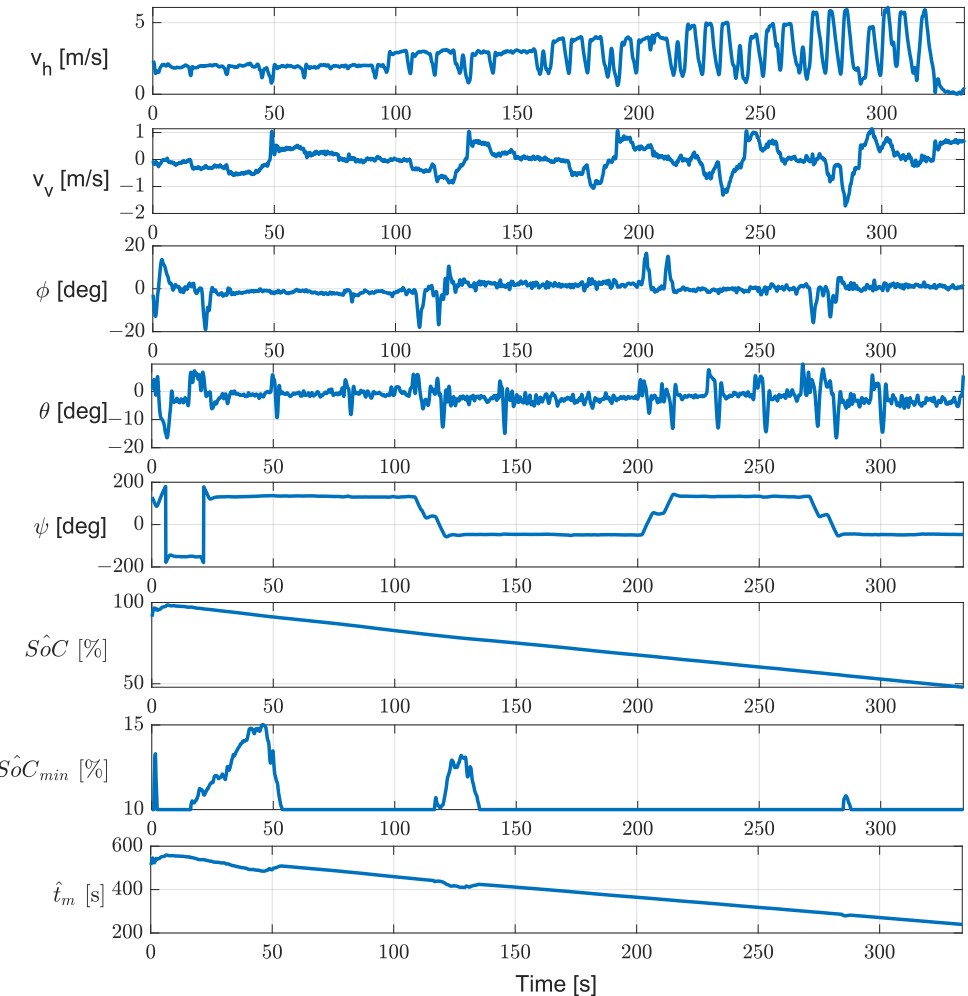

**Figure 17.** Parameters obtained from the validation flight. From top to bottom, horizontal velocity ($v_v$), vertical velocity ($v_h$); roll ($\phi$), pitch ($\theta$), yaw ($\psi$) angles; estimated state of charge, $S\hat{o}C$; minimum admissible state of charge, $S\hat{o}C_{min}$; and flight time margin, $\hat{t}_m$.

## 5. Conclusions and Future Work

In this research work, a new configuration for the estimation of the energy required to perform a multirotor UAV mission using fuzzy c-means was presented. This estimator was able to make predictions of the required energy with a maximum error equivalent to the energy required to fly for 7 s.

The optimization process of the clustering parameters using PSO allowed us to determine the optimal value for the weighting exponent $m$ and the number of clusters for each subsystem. Although the proposed methodology only considers the multirotor UAV's velocities, for which the use of two clusters per sub-system was determined, this architecture can be extended to include other parameters that affect the energy consumption of the vehicle without modifying the overall system structure, and we employed the proposed PSO-based methodology to determine the optimal system parameters.

The relationship between the state of charge of the battery and the thrust control signal was analyzed, and a methodology was presented to determine the minimum admissible charge level to operate the multirotor UAV safely. This method can be especially useful when the vehicle is operated outside the design conditions, as in the case of a failure during a mission or in environments different from those considered in its design.

With respect to the way to determine the state of charge, although the EKF-based methodology has been widely used, its use was proposed in conjunction with T-S fuzzy models for the relationship between $V_{oc}$ and the $SoC$, which allows combining the simplicity of linear functions with the smoothness of transitions between function segments.

Since energy estimation is a complex multiparameter-dependent problem, it is required to extend the number of input parameters to include factors such as the payload weight, operating altitude, wind speed, and relative wind direction. To generate the training set, a combination of experimental flight data and high-precision simulations were proposed so that the system could determine the energy required in situations where the vehicle has not been exposed.

It is proposed to evaluate new meta-heuristic algorithms for the optimization of the parameters of the energy estimation system, which can offer more effective alternatives in terms of execution time, which is essential if a greater number of parameters for energy estimation are to be added.

Regarding the battery parameters, it is proposed to incorporate the effect of battery health and temperature into the model to increase the accuracy of the $SoC$ estimation, and consequently, of the flight time margin.

**Author Contributions:** Conceptualization, R.L. and E.S.E.; methodology, L.H.M. and E.S.E.; software, L.H.M. and J.C.R.-F.; validation, L.H.M. and J.C.R.-F.; formal analysis, L.H.M. and E.S.E.; investigation, L.H.M. and E.S.E.; resources, R.L. and E.S.E.; writing—original draft preparation, L.H.M. and E.S.E.; writing—review and editing, L.H.M., E.S.E., and J.C.R.-F.; visualization, L.H.M. and J.C.R.-F.; supervision, E.S.E. and R.L.; project administration, E.S.E. and R.L.; funding acquisition, E.S.E. and R.L. All authors have read and agreed to the published version of the manuscript.

**Funding:** This work was partially supported by the Mexican National Council for Science and Technology through Project 321224: National Laboratory of Autonomous Vehicles and Exoskeletons

**Institutional Review Board Statement:** Not applicable.

**Informed Consent Statement:** Not applicable.

**Data Availability Statement:** Not applicable.

**Conflicts of Interest:** The authors declare no conflict of interest.

## Abbreviations

The following abbreviations are used in this manuscript:

| Acronyms | Description |
|---|---|
| EKF | Extended Kalman Filter. |
| MAF | Moving Average Filter. |
| PSO | Particle Swarm Optimization. |
| RLS | Recursive Least Squares. |
| SoC | State of Charge. |
| T-S | Takagi–Sugeno. |
| UAV | Unmanned Aerial Vehicle. |
| **Fuzzy system variables** | **Description** |
| $x, x_k$ | Input to the fuzzy system. |
| $A_i$ | Fuzzy set. |
| $y_i$ | Output of the fuzzy rule. |
| $a_i, b_i$ | Design parameters of the consequent fuzzy rule. |
| $M$ | Number of rules of the fuzzy system. |
| $u_i$ | Membership degree of the input $x$ to the rule. |
| $a, b, c, d$ | Trapezoidal membership function characteristic points. |
| $N$ | Number of elements of the data set. |
| $J_m$ | Cost function of the fuzzy C-means algorithm. |
| $v$ | Center vector of the cluster. |
| $m$ | Weighting exponent for fuzzy C-means. |
| $X$ | Set of training input data. |
| $U_i$ | Set of membership values of the i-th rule for the set of training data set. |
| $X_e, X'$ | Auxiliary matrices for calculating of T-S consequent. parameters. |
| $\theta$ | Vector of parameters of the T-S rules. |
| **PSO algorithm variables** | **Description** |
| $J$ | Function to minimize by the PSO algorithm. |
| $p_i$ | Position of the i-th particle. |
| $p_{b_i}$ | Best local of the i-th particle. |
| $p_{b_g}$ | Best global among all particles. |
| $v_i$ | Velocity of the i-th particle. |
| $\chi$ | Velocity constrain factor of the particle. |
| $c_1$ | Weighting factor of the cognitive component. |
| $c_2$ | Weighting factor of the social component. |
| $\phi$ | Auxiliary variable for the calculation of $\chi$. |
| **Energy estimation system variables** | **Description** |
| $P_i$ | Measured power consumption at current state. |
| $\hat{P}$ | Estimated required power at current state. |
| $e_P$ | Estimated required power error. |
| $v_v$ | Vertical velocity. |
| $v_h$ | Horizontal velocity. |
| $V_{v_m}$ | Set of vertical velocities of the mission profile. |
| $V_{h_m}$ | Set of horizontal velocities of the mission profile. |
| $T_i$ | Period during the UAV moves at a given velocity. |
| $M'_i$ | Auxiliary variable to determine the number of clusters. |
| $\tilde{e}_p$ | Output of the MAF with input $e_p$. |
| **Battery equivalent circuit variables** | **Description** |
| $C_T$ | Capacitance to model the battery capacity. |
| $C_d$ | Capacitance to model battery transient. |
| $R_i$ | Internal battery resistance. |
| $R_d$ | Resistance to model battery transient. |
| $Q_B$ | Energy stored in the battery. |

| | |
|---|---|
| $\eta$ | Efficiency factor of the battery. |
| $I_b$ | Current through battery terminals. |
| $V_b$ | Voltage on battery terminals. |
| $V_d$ | Battery transient voltage. |
| $V_{oc}$ | Open circuit battery voltage. |
| Flight time margin variables | Description |
| $\xi_d$ | Position reference. |
| $\omega_d$ | Angular velocity reference. |
| $\delta_T$ | Thrust control signal. |
| $\delta_{T_{max}}$ | Maximum admissible value of $\delta_T$. |
| $\delta_p, \delta_q, \delta_r$ | Control signals for angular velocity . |
| $k_f$ | Force constant of the rotor. |
| $k_\omega$ | Angular velocity constant of the rotor. |
| $u_d$ | Duty cycle of the control signal. |
| $V_m$ | Average voltage applied to the motor. |
| $V_{b_{nom}}$ | Nominal voltage of the battery. |
| $V_{b_\delta}$ | Battery voltage for a given $\delta_t$. |
| $V_{oc_{min}}$ | Open circuit voltage at minimum admissible *SoC*. |
| $V_{b_s}$ | Minimum admissible voltage at the battery terminals defined by the operator. |
| $V_{b_{min}}$ | Minimum voltage at the battery terminals at which it is possible to operate the UAV. |
| $P_m$ | Average required power. |
| $SoC_s$ | Minimum charge of the battery defined by the operator. |
| $S\hat{o}C_{min}$ | Minimum battery charge at which it is possible to operate the UAV. |
| $S\hat{o}C_0$ | Estimated *SoC* at the time of flight time evaluation. |
| $S\hat{o}C_f$ | Expected *SoC* at the end of the mission. |
| $\hat{E}_R$ | Estimated required energy to mission accomplishment. |
| $E_T$ | Energy stored in the battery when fully charged. |
| $e_{E_R}$ | Required energy estimation error. |
| $\hat{t}_m$ | Estimated flight time margin. |

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
