# Peer review of "Estimation of Energy Consumption and Flight Time Margin for a UAV Mission Based on Fuzzy Systems"

_technologies, doi:10.3390/technologies11010012_

Round 1

Reviewer 1 Report

The proposed work is fantastic. Please modify this manuscript as per the following suggestions:

1. Why Kalman filter was used to estimate the remaining charge? Since "n" number of advanced filters are available than KF, why has this KF been imposed for this work? 

2. Since the experiments were conducted on hexarotor UAV, I can recommend incorporating "multirotor UAV" instead of "UAV" in prominent places such as keywords, titles and abstract.  

3. I have calculated that your complete endurance is 350 s, so please provide the mission profile of your focused UAV. This kind of mission profile incorporation may additionally create viewability for this manuscript. 

4. From the literature survey section, it is observed that the previously completed works are relevant to the fuzzy imposition of not being incorporated properly. Thus, I recommend this team add additional literature relevant to fuzzy imposed on drones. 

5. Many abbreviations, such as PSO, SoC, etc., are not expanded. Please do the needful in this regard. Also, I advise you to check this error throughout the main text.

6. "From the defined rules, the value of ˆP is calculated using Equation (5)." Is it the correct equation number?

7. Use UAV or unmanned aircraft sensors instead of aircraft sensors. Similarly, please adjust the necessary corrections. 

8. Is MAF the same as KF or something else? 

9. In line 260, the author mentioned that "of the state of charge using an Extended Kalman filter is addressed." Is it EKF or KF? If it is EKF, why in the abstract are the authors mentioning the importance of KF? Please justify your answer. 

10. In line 304, it is mentioned that "The four control signals obtained are used by a control allocation system". is it four or six? 

Reviewer 2 Report

The authors mention that there are two methods (mathematical and empirical models) for required energy estimation. What are the advantages of the fuzzy logic and PSO-based method you propose over these two methods? It should be explained in the introduction section.

Why did you select Takagi-Sugeno fuzzy for energy estimation for the UAV? And Why did you select the PSO algorithm? Could the newly introduced meta-heuristic algorithms in literature also provide good performance than PSO?

A nomenclature table can be added to the article to define the parameters in your formula.

As far as I understand, battery degradation is not taken into account in your energy estimation model. Why? Can be it add?

What is the objective function used for the PSO algorithm? what is the number of the equation?

A comparison with the literature for the method you propose should be added to the results and discussion.

In figure 16, why is so much difference between the two signals?

The future works recommendation can be extended.

Reviewer 3 Report

Dear Authors,

please, refer to the attached document. Thanks!

Reviewer 4 Report

the abstract does not describe the content. This part needs to be written again. It does not contain a goal, it does not specify the methodology or the overall results achieved. The article is written very correctly. I miss research questions and hypotheses. Indicated study limitations.

Round 2

Reviewer 1 Report

Please refer to the previously published article and so create a good mission profile picture that will enhance your viewability. 

Reviewer 3 Report

Dear Author,

please refer to the attached document. Thanks!
